# FedBEns: One-Shot Federated Learning based on Bayesian Ensemble

**Jacopo Talpini** [1]   **Marco Savi** [1]   **Giovanni Neglia** [2]

## Abstract

One-Shot Federated Learning (FL) is a recent paradigm that enables multiple clients to cooperatively learn a global model in a single round of communication with a central server. In this paper, we analyze the One-Shot FL problem through the lens of Bayesian inference and propose FedBEns, an algorithm that leverages the inherent multimodality of local loss functions to find better global models. Our algorithm leverages a mixture of Laplace approximations for the clients' local posteriors, which the server then aggregates to infer the global model. We conduct extensive experiments on various datasets, demonstrating that the proposed method outperforms competing baselines that typically rely on unimodal approximations of the local losses.

## 1. Introduction

Federated Learning (FL) is a collaborative machine learning paradigm that has recently gained significant attention from the research community. It enables the training of models using decentralized datasets stored locally on multiple clients, without requiring direct access to the raw data (McMahan et al., 2017; Kairouz et al., 2021). This paradigm is particularly appealing in scenarios involving highly sensitive data, where clients are reluctant to share them, due to privacy or security concerns. Most FL algorithms typically train models using iterative optimization techniques that require numerous rounds of communication between the clients and the central server that aggregates the local models.

The frequent server-client interactions represent a bottleneck in FL (Guha et al., 2019; Kairouz et al., 2021), which has been addressed by a new paradigm called One-Shot FL, where the global model is learned in a single round of communication between clients and the server (Guha et al., 2019).

Several One-Shot FL algorithms have been proposed in the literature. Existing relevant work leverages knowledge distillation at the server (Lin et al., 2020), neuron matching strategies (Singh & Jaggi, 2020) or adopts an optimization approach, trying to directly approximate the global loss at the server starting from the local losses of each client (Jhunjhunwala et al., 2024; Liu et al., 2024; Matena & Raffel, 2022). Our contribution is in line with the last group of work, which generally employs a unimodal approximation of each local loss. As an example, Jhunjhunwala et al. (2024) make use of a quadratic approximation of local losses through opportune Fisher information matrices.

However, neural network loss functions are known to be highly non-convex (Izmailov et al., 2021; Wilson & Izmailov, 2020; Fort et al., 2019; Lakshminarayanan et al., 2017) with multiple local minima, some of which may generalize better than others. This fact can be extremely relevant for the considered One-Shot FL setting and motivated us to introduce a novel algorithm that explicitly takes into account the different modes of the loss surface, guided by a Bayesian analysis of the FL problem.

Figure 1 shows the importance of considering multiple modes, motivating our proposed approach. The figure illustrates that the local minimum corresponding to the broader, less sharply-peaked mode of client 1's local loss, is crucial for accurately reconstructing the optimal global model. Moreover, it is worth noting that this represents a relatively simple scenario where the global loss is unimodal. In general, however, the global loss exhibits a complex landscape with multiple optima, and selecting just one of them as global model may lead to suboptimal results (Wilson & Izmailov, 2020). We argue that this consideration is especially crucial for learning a model in a One-Shot setting while achieving good predictive performance, as we will show in the rest of the paper. The key insight, well-established in Bayesian inference (MacKay, 2003), is that describing the whole posterior distribution should be the main goal of learning from data, while a local description around the posterior's maximum may not be sufficiently informative.

**Contributions.** In this paper, we introduce FedBEns (Federated Bayesian Ensemble), a straightforward and principled

---

[1]Department of Informatics, Systems and Communication (DISCo), University of Milano-Bicocca, Milan, Italy; [2]Inria, Université Côte d'Azur, Sophia Antipolis, France. Correspondence to: Jacopo Talpini <j.talpini@campus.unimib.it>.

*Proceedings of the 42^nd International Conference on Machine Learning*, Vancouver, Canada. PMLR 267, 2025. Copyright 2025 by the author(s).

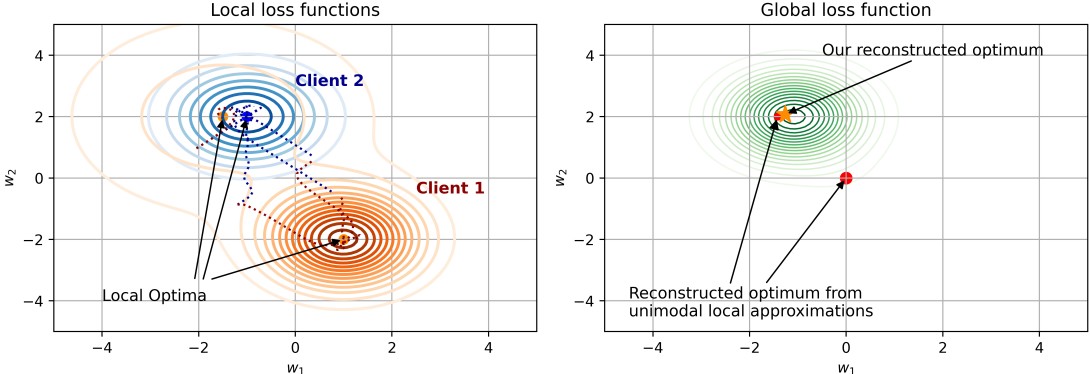

*Figure 1.* An illustration of One-Shot FL in a toy 2D setting with two clients: one with a unimodal loss function (Client 2, left plot) and the other with a bimodal loss function (Client 1, left plot), along with the overall 'ground-truth' global loss (right plot). The left plot also shows several stochastic gradient descent (SGD) trajectories on the two losses (dotted lines), to mimic clients' training, starting from different random initializations. In the right plot, the red dots represent the reconstructed global optimum, depending on whether client 1 approximates its local loss as quadratic around its global optimum (a more likely result, represented by the bigger dot) or around the secondary minimum. The yellow star denotes the global optimum inferred by our approach, based on an ensemble of all the optimal solutions found by each SGD run. To calculate the global loss, the secondary minimum of Client 1 is more relevant than its global minimum.

one-shot federated learning method based on Bayesian analysis of the distributed inference problem. We derive an exact expression to combine clients' local model posteriors, yielding the same global posterior one could compute from the union of all local datasets. Guided by this result, we propose a practical algorithm based on a multimodal approximation of the local loss functions—and, consequently, the global loss function—constructed using a mixture of local Laplace approximations. We subsequently exploit this multimodal global posterior to approximate Bayesian marginalization, rather than relying solely on its optimum, to derive the global model. Finally, we empirically evaluate the performance of our method on standard benchmarks and we compare it with several baselines and state-of-the-art approaches. Our illustrative numerical results demonstrate that our proposal outperforms other state-of-the-art models across different scenarios.

The remainder of the paper is organized as follows. In Section 2 we revise several relevant related works, while in Section 3 we analyze the One-Shot FL problem from a probabilistic perspective and we describe the proposed approach in Section 4. In Section 5 we describe the experimental setup, the benchmark approaches from the literature, and the considered datasets. In Section 6 we provide and discuss the numerical results, and conclude the paper by highlighting the main takeaways and lessons learned in Section 7.

## 2. Related Work

In this Section, we begin by reviewing seminal work on Bayesian approaches for Federated Learning, followed by an in-depth discussion of One-Shot FL.

### 2.1. Bayesian approaches to Federated Learning

Federated Learning is typically framed as a distributed optimization problem characterized by unique challenges, including unbalanced and non-i.i.d. data distributions across clients and constraints on communication. A seminal contribution that integrates Bayesian principles and FL is the FedPA algorithm (Al-Shedivat et al., 2021). This algorithm requires multiple communications rounds and it is based on Markov Chain Monte Carlo for approximate inference of local posteriors on the clients. It efficiently communicates their statistics to the server, which uses them to refine a global estimate of the posterior mode. A further contribution is FedEP (Guo et al., 2023), which exploits an expectation propagation strategy, where approximations to the global posterior are iteratively refined through probabilistic message-passing between the central server and the clients. For a systematic review of FL with Bayesian models see (Cao et al., 2023).

Although the theoretical foundations of previous works are rooted in Bayesian theory, their primary goal is to infer the maximum of the posterior, achieved through multiple rounds of communication. In contrast, our focus is on combining models in a single round of communication and relying on Bayesian marginalization.

## 2.2. One-Shot Federated Learning

Existing work for One-Shot FL can be broadly split into three main categories: (i) knowledge distillation methods, (ii) neuron matching methods and (iii) loss-based models fusion.

The methods falling in the first group view the client models as an ensemble and the overall idea is to distill the knowledge from this ensemble into a single global model. To accomplish that, some works assume the existence of a dataset at the server (Gong et al., 2021; Lin et al., 2020; Guha et al., 2019) while others mimic this scenario by training generative models, like GAN or VAE, to synthesize an auxiliary dataset at the server side (Zhang et al., 2020; Zhou et al., 2020; Heinbaugh et al., 2023). In another recent paper, Hasan et al. (2024) propose an approach based on a combination of predictive mixture models and a Bayesian committee machine. The contribution of each component is weighted by a parameter, which is determined at the server by minimizing the log-likelihood using a server-side dataset, also used to distill the committee's knowledge into a single model. The availability of a dataset (whether real or synthetic) raises several concerns. If each client shares a portion of its data, privacy issues arise. Alternatively, a datasets obtained through other means may not accurately represent the clients' data. Additionally, using such a dataset increases the computation cost on the server and requires careful hyperparameter tuning, posing a significang challenge for implementation in One-Shot FL (Jhunjhunwala et al., 2024).

On the other hand, methods based on neuron matching rely on the invariance of Neural Networks (NN) w.r.t. permutations of the neurons. Some works leverage this property, by first aligning the weights of the client models according to a common ordering (called matching) and then averaging the aligned client models (e.g. (Singh & Jaggi, 2020; Liu et al., 2022; Jordan et al., 2023)). Although this approach has been shown to work well when combining simple models like feedforward NNs, its performance declines when combining more complex ones such as CNNs (Jhunjhunwala et al., 2024).

The last group of works, referred to as model fusion, intends to fuse the capabilities of multiple existing models into a single model (Yadav et al., 2023; Jin et al., 2023; Matena & Raffel, 2022). These approaches were not specifically designed for one-shot federated learning. As a result, they overlook the fact that the models to be aggregated may be trained on datasets drawn from different distributions, as is common in federated learning. A relevant contribution to the field, more aligned with our work, is (Jhunjhunwala et al., 2024). Here, the authors propose a One-Shot FL approach based on the Fisher information matrix. Each client computes a local estimate of the model weights by minimizing

the local loss function and evaluating its curvature around the minimum. This information is then transmitted to the server. The server reconstructs quadratic approximations of the local loss functions from the received information and employs stochastic gradient descent (SGD) to minimize the sum of these approximations. A regularization term is included in the server's loss function to enhance stability by preventing SGD from drifting too far from the mean of the local model parameters. Our approach aligns with this last group of works, but it differs by providing a multimodal approximation of the loss functions and by relying on the marginalization of the posterior rather than its optimization, thus embracing a full-Bayesian approach (Wilson & Izmailov, 2020).

## 3. Problem Formulation

The One-Shot FL problem can be formulated as follows: given a collection of $C$ clients, where each client $c$ has its own dataset of input-outputs pairs $\mathcal{D}_c = \{(\mathbf{x}_i, \mathbf{y}_i)\}_{i=1}^{N_c}$, the goal is to learn a parametric function (e.g., a Neural Network) that provides the conditional probability distribution $p(\mathbf{y}|\mathbf{x}, \mathbf{w})$, for a given input $\mathbf{x}$ and model parameters $\mathbf{w}$ (Murphy, 2023), in a distributed way. Specifically, we aim to combine the local models trained on each $\mathcal{D}_c$ such that the resulting global model is equivalent to the one trained on $\mathcal{D} = \cup_c \mathcal{D}_c$. Local models are trained by minimizing a specified loss function, which typically corresponds to the negative log-likelihood induced by the considered probabilistic model $p(\mathbf{y}|\mathbf{x}, \mathbf{w})$ (Al-Shedivat et al., 2021). For example, the commonly used least-squares loss for regression can be derived from the likelihood under a Gaussian model, while the cross-entropy loss corresponds to the likelihood under a categorical model. Overall, the local learning process at each client $c$ is framed as the minimization of a given loss function $\mathcal{L}_c(\mathbf{w})$.

### 3.1. Combining models: a probabilistic perspective

A principled way to combine the local models is through the likelihood $p(\mathcal{D}|\mathbf{w}) = p(\mathcal{D}_1, .., \mathcal{D}_C|\mathbf{w})$. Assuming that the clients' data are independent given the model, the global likelihood can be factorized into a product of local likelihoods as follows:

$$p(\mathcal{D}_1, .., \mathcal{D}_C|\mathbf{w}) = \prod_{c=1}^{C} p(\mathcal{D}_c|\mathbf{w}) \tag{1}$$

To obtain the global posterior, instead of combining the likelihoods, it is possible to combine directly the client's posteriors, as specified by the following Proposition:

**Proposition 3.1.** *Given $C$ datasets and the posteriors $p(\mathbf{w}|\mathcal{D}_c)$ of the same model $f(\mathbf{w})$, under the assumptions that (i) the datasets are conditionally independent given*

*the model, and (ii) the prior is the same across clients, the global posterior can be written as:*

$$p(\mathbf{w}|\mathcal{D}_1, .., \mathcal{D}_C) = \alpha \cdot \frac{\prod_{c=1}^{C} p(\mathbf{w}|\mathcal{D}_c)}{p(\mathbf{w})^{C-1}}. \tag{2}$$

The first term of the right-hand side is a normalizing constant and does not depend explicitly on the model parameters, while the factor $p(\mathbf{w})^{C-1}$ avoids the over-counting of the prior. A proof is given in the Appendix. We emphasize that this result allows for the exact combination of distributed inference outcomes represented by the client posteriors $\{p(\mathbf{w}|\mathcal{D}_c)\}_{c=1}^{C}$, without any loss of information. In other words, combining the posteriors via Equation (2) provides the same information about the model as if it had been trained in the usual centralized setting, by combining directly the clients' data, thus naturally enabling Federated Learning in One-Shot.

If we are interested in finding the maximum of the global posterior, i.e., to perform a Maximum a Posteriori (MAP) estimate of the parameters for the global model, we can rely on the log-posterior and drop the terms that do not depend on $\mathbf{w}$, resulting in the following objective function:

$$\mathcal{L}(\mathbf{w}) = \sum_{c=1}^{C} \log(p(\mathbf{w}|\mathcal{D}_c)) - (C-1)\log(p(\mathbf{w})) \tag{3}$$

**Remark.** It is worth mentioning that under the assumption of an (improper) flat prior the obtained result reduces to what has been presented in (Al-Shedivat et al., 2021), and considered as starting point in (Jhunjhunwala et al., 2024). While those works focus solely on inferring the maximum of the posterior, providing a unimodal approximation of the local loss, we propose a multimodal method that properly accounts for the prior by capturing its different modes.

## 4. Proposed Algorithm: FedBEns

In this Section we discuss in detail our approach, which operates in two stages. In the first stage, the global posterior is estimated, followed by a second phase where its different modes are identified. The details of our method are given in the pseudocode presented in Algorithm 1.

### 4.1. Global posterior estimation

As mentioned, we first aim to construct a tractable and effective approximation of the complex local posteriors that can capture different modes. To achieve this, we employ a mixture of Gaussian distributions to approximate the local posterior distribution of each client, $p(\mathbf{w}|\mathcal{D}_c)$. More specifically, we exploit a mixture of Laplace approximations (LA) (MacKay, 1992) of independently-trained Deep Neural Networks (Eschenhagen et al., 2021). The rationale behind

this choice lies in combining a lightweight approach to describe the local curvature of the loss (provided by the usual Laplace approximation) with a global perspective provided by the different ensembles (Eschenhagen et al., 2021). More concretely, each client trains $M$ times the same model on the same data but randomly varying the starting point of the gradient descent algorithm, exploited for the training. This approach is advantageous as it does not require any modifications to the model architecture, the standard loss function, or the learning process. By adopting this method, it follows that the overall posterior at the server will be a mixture of Gaussian distributions and, in general, multimodal. More precisely, starting from Equation (2), the posterior for $C$ clients and $M$ mixtures can then be written as:

$$p(\mathbf{w}|\hat{\mathbf{w}}_{1:M}^{1:C}, \Lambda_{1:M}^{1:C}) \propto$$
$$\frac{1}{p(\mathbf{w})^{C-1}} \prod_{c=1}^{C} \sum_{m=1}^{M} \frac{1}{M} \mathcal{N}(\mathbf{w}|\hat{\mathbf{w}}_m^c, \Lambda_m^c) \tag{4}$$

where: $\hat{\mathbf{w}}_m^c$ is the $m$-th maximum-a-posteriori estimate of the $c$-th client, found through standard gradient-based optimizers, and similarly $\Lambda_m^c$ is the precision matrix. The precision matrix involved in the LA is derived from the Hessian of the negative log-likelihood evaluated at $\hat{\mathbf{w}}_m^c$. To ensure positive definiteness, it is typically approximated using the generalized Gauss-Newton method (Daxberger et al., 2021a). The previous expression is used by the server as a proxy for the global posterior. Specifically, to identify different modes of this posterior, used in the second phase of our approach, the server computes the log-posterior, thus obtaining the following objective function:

$$\mathcal{L}(\mathbf{w}|\hat{\mathbf{w}}_{1:M}^{1:C}, \Lambda_{1:M}^{1:C}) = \sum_{c=1}^{C} \log(\sum_{m=1}^{M} \frac{1}{M} \mathcal{N}(\mathbf{w}|\hat{\mathbf{w}}_m^c, \Lambda_m^c)) +$$
$$- (C-1)\log(p(\mathbf{w})) \tag{5}$$

**Remark.** In the previous equations, we implicitly assumed that each mixture component has equal weight. This assumption aligns with (Eschenhagen et al., 2021), as we do not expect any particular model to be more important than the others, given that the only difference between the mixture components is their random initialization.

### 4.2. Model predictions

Given the posterior distribution, a fully-Bayesian approach makes predictions by relying on marginalization over the posterior of the model parameters, rather then using a single setting of weights (Wilson & Izmailov, 2020). More specifically, by marginalizing, one computes the predictive distribution for a given input $\mathbf{x}$ as (Wilson & Izmailov, 2020; MacKay, 1995):

$$p(\mathbf{y}|\mathbf{x}, \mathcal{D}) = \int p(\mathbf{y}|\mathbf{x}, \mathbf{w})p(\mathbf{w}|\mathcal{D})d\mathbf{w} \tag{6}$$

As mentioned earlier, the posterior is generally multimodal, indicating that it is possible to find different parameter settings that can fit the data well. Therefore, it is advantageous to consider a mixture of these solutions and use all of them to make predictions, in line with a Bayesian Model Averaging perspective, represented by Equation (6) (Wilson & Izmailov, 2020).

---

**Algorithm 1** FedBEns: One-Shot FL through Federated Bayesian Ensemble

**Require:** Data $\mathcal{D}_{1:C}$ (data of each client); number of mixtures $M$; client and server learning rates $\eta_c, \eta_s$; and number of iterations $K_c, K_s$, respectively.
1: **Server** randomly initializes $M$ model weights $\mathbf{w}^0_{1:M}$ and broadcast them to $C$ selected clients
2: **for** each client $c$ in $C$ **in parallel do**
3:    $\hat{\mathbf{w}}^{(c)}_{1:M}, \Lambda^{(c)}_{1:M} \leftarrow \texttt{ClientTraining}(\mathbf{w}^0_{1:M}, \mathcal{D}_c, K_c)$
4:    **Client** $c$ sends updated mixture weights and precision matrices $\hat{\mathbf{w}}^{(c)}_{1:M}, \Lambda^{(c)}_{1:M}$ to the server
5: **end for**
6: **Server** aggregates client updates to infer global optima:
7: $\hat{\mathbf{w}}^{\text{global}}_{1:M} \leftarrow \texttt{ServerAggregation}(\hat{\mathbf{w}}^{1:C}_{1:M}, \Lambda^{1:C}_{1:M})$
8: **Clients** receive $\hat{\mathbf{w}}^{\text{global}}_{1:M}$, and make predictions by ensembling:
9:    $p(\boldsymbol{y}|\mathbf{x}, \mathcal{D}_{1:C}) = \frac{1}{M} \sum_{m=1}^{M} p(\boldsymbol{y}|\mathbf{x}, \hat{\mathbf{w}}^{\text{global}}_m)$

10: **Function:** $\texttt{ClientTraining}(\mathbf{w}^0_{1:M}, \mathcal{D}_c, K_c)$
11: **for** each mixture model $m$ **do**
12:    **for** $k = 0, \ldots, K_c - 1$ **do**
13:       $\mathbf{w}^{k+1}_m \leftarrow \mathbf{w}^k_m - \eta_c \sum_i \nabla \ell(\mathbf{w}^k_m; \mathbf{x}_i, y_i)$
14:    **end for**
15:    Set $\hat{\mathbf{w}}_m = \mathbf{w}^{K_c-1}_m$
16:    Compute precision matrix of Laplace approx. $\Lambda_m$
17: **end for**
18: **return** $\hat{\mathbf{w}}_{1:M}, \Lambda_{1:M}$
19: **Function:** $\texttt{ServerAggregation}(\hat{\mathbf{w}}^{1:C}_{1:M}, \Lambda^{1:C}_{1:M}, K_s)$
20: Compute global loss $\mathcal{L}(\boldsymbol{w}|\hat{\mathbf{w}}^{1:C}_{1:M}, \Lambda^{1:C}_{1:M})$ via Eq. (5)
21: Compute starting points: $\boldsymbol{w}^0_{1:M} = \text{median}_c(\hat{\mathbf{w}}^{1:C}_{1:M})$
22: **for** each mixture model $m$ **do**
23:    **for** $k = 0, \ldots, K_s - 1$ **do**
24:       $\mathbf{w}^{k+1}_m \leftarrow \mathbf{w}^k_m - \eta_s \nabla \mathcal{L}(\mathbf{w}^k_m)$
25:    **end for**
26:    Set $\hat{\mathbf{w}}^{\text{global}}_m = \mathbf{w}^{K_s-1}_m$
27: **end for**
28: **return** $\hat{\mathbf{w}}^{\text{global}}_{1:M}$

---

A convenient way to approximate Equation (6) is again through ensembling so that the predictive distribution is approximated as $p(\boldsymbol{y}|\mathbf{x}, \mathcal{D}) \sim \frac{1}{M} \sum_{m=1}^{M} p(\boldsymbol{y}|\mathbf{x}, \hat{\mathbf{w}}_m)$, where $\hat{\mathbf{w}}_m$ represent a mode of the posterior (Wilson & Izmailov, 2020; Murphy, 2023). To find different modes of the global posterior, the server adopts the same approach as each client, by running SGD $M$ times on the approximate global log-

posterior (Lakshminarayanan et al., 2017), initializing from the median of the different MAP solutions obtained from the clients' mixtures. This ensemble represents the final model that the server sends back to the clients.

## 4.3. Implementation considerations

The Laplace approximation involves computing the Hessian at the MAP estimate. However, constructing the full Hessian matrix is quadratic in the number of parameters, making it impractical for modern Neural Networks. Typical approaches for applying the Laplace approximation include diagonal or Kronecker factorization of the Hessian (Grosse & Martens, 2016; Ritter et al., 2018), or applying the Laplace approximation to a subset of the model (e.g., to the last layers) (Daxberger et al., 2021a). Here, we consider two approaches for approximating the Hessian:

**Diagonal+Full**. This approach approximates the Hessian simply with its diagonal elements, with the exception of the last layer for which it exploits the full Hessian. This resembles the subnetwork LA (Daxberger et al., 2021b; Kristiadi et al., 2020) that treats only a subset of the model parameters probabilistically, typically focusing on the last linear layer while ignoring the uncertainty associated with the feature extractor layers. Last-layer Laplace approximations proved to be effective in practice, in the usual centralized setting (Daxberger et al., 2021a).

**Kronecker**. More expressive alternatives to the simple diagonal approximation are block-diagonal factorizations, such as Kronecker-factored approximate curvature (Martens & Grosse, 2015), which factorizes the Hessian of each layer as a Kronecker product of two smaller matrices. The dimensions of these matrices are determined by the number of input-output neurons. Let $d_1$ and $d_2$ denote the number of input and output neurons, respectively. Consequently, the computational cost for storing and communicating the Hessian scales as $\mathcal{O}(d_1^2 + d_2^2)$, instead of $\mathcal{O}((d_1 \cdot d_2)^2)$.

**Temperature Scaling.** In Bayesian deep learning it is common to consider a tempered posterior by raising the likelihood to the power $1/T$: $p_{temp}(\boldsymbol{w}|\mathcal{D}) \propto p(D|\boldsymbol{w})^{1/T} p(\boldsymbol{w})$ (Wilson & Izmailov, 2020). Smaller values for the temperature (i.e., $T < 1$) lead to more concentrated posteriors. A low temperature is beneficial since it can improve our approximation of the true Laplace by reducing the overestimation of variance in certain directions, which arises when covariances between layers are ignored (Ritter et al., 2018). Additionally, it enhances the LA itself by preventing it from placing too much probability mass in low-probability regions of the true posterior (Ritter et al., 2018).

**Choice of the Prior.** We exploited the commonly-used zero-mean Gaussian diagonal prior, $p(\mathbf{w}) = \mathcal{N}(0, \sigma^2 I)$, where $\sigma^2$ is the variance (Wilson & Izmailov, 2020).

# 5. Experimental Setup

## 5.1. Benchmark methods

We compared our method with well-established state-of-the-art approaches for one-shot federated learning, ensuring that we included representative algorithms from each category described in Sec. 2.2, with a special focus on methods that do not explicitly require a dataset to be available at the server or involve multiple server-client communications.

**FedFisher** (Jhunjhunwala et al., 2024). As we discussed in Sec. 2.2, this strategy is based on an unimodal approximation of the loss as a quadratic function. The Fisher matrix is approximated through a diagonal matrix or through a Kronecker factorization. Here, we consider for comparison the method based on Kronecker approximation, referred to as FedFisher K-FAC, which achieves better results.

**RegMean** (Jin et al., 2023). It is a model fusion approach that merges models in their parameter space, guided by weights that minimize prediction differences between the merged model and the individual models.

**DENSE** (Zhang et al., 2022). It is a strategy that relies on data-free knowledge distillation, by exploiting GANs to artificially generate data for distillation at the server.

**OTFusion** (Singh & Jaggi, 2020). It is a popular neuron-matching method that utilizes optimal transport to align neurons across the models by averaging their associated parameters, rather than directly averaging the weights.

**Fisher Merge** (Matena & Raffel, 2022). It is a model fusion baseline that, similarly to (Jhunjhunwala et al., 2024), employs a diagonal approximation to the Hessian of the log-posterior.

## 5.2. Datasets and models

To assess the performance of the proposed strategy, and to compare it with other baselines, we conducted several experiments on standard benchmarks and settings common in the literature (see for example (Jhunjhunwala et al., 2024)). We considered the following datasets:

**FashionMNIST** (Xiao et al., 2017). It is a dataset of Zalando's article images, consisting of a training set of $60,000$ samples and a test set of $10,000$ instances. Each sample is a $28 \times 28$ grayscale image, associated with a label from 10 classes. Here we refer to this dataset as FMNIST.

**SVHN** (Netzer et al., 2011). It is a real-world image dataset obtained from house numbers in Google Street View images. It consists of $600,000$ digit images associated with a label from 10 classes.

**CIFAR10** (Krizhevsky et al., 2010). It consists of $60,000$ $32 \times 32$ color images in 10 different classes (e.g. airplane, dog, truck, etc.), with $6,000$ images per class. There are $50,000$ training images and $10,000$ test images.

To simulate data heterogeneity among $C$ client datasets, we partitioned the original image dataset into $C$ subsets using a symmetric Dirichlet sampling procedure with parameter $\alpha$ (Hsu et al., 2019), where a smaller value of $\alpha$ results in a more heterogeneous data split. In addition, data are normalized before splitting, and a small fraction of the training data (i.e., 500 samples) is kept at the server as validation data for the hyperparameter tuning of the various approaches.

Regarding the models employed for the classification task, we used two CNNs as done in the literature (Jhunjhunwala et al., 2024): LeNet (LeCun et al., 1998) for FashionMNIST, and a larger CNN proposed by (Wang et al., 2020) for the other two datasets.

## 5.3. Training details

We kept the basic training procedure fixed for all the approaches and experiments to compare them fairly. SGD is utilized as local optimizer, with each client training for 20 epochs on LeNet and 50 epochs on the more complex CNNs for SVHN and CIFAR10. Moreover, we set the batch size to 64, the learning rate $\eta_c$ to 0.01, and the momentum to 0.9, in line with (Jhunjhunwala et al., 2024). For the various parameters of each baseline, we used their default settings and the official implementations provided by the authors.

Concerning FedBEns, we used the standard PyTorch weights initializer for the random initialization of the local ensemble. To perform Laplace approximation of the local posteriors, we exploited the open-source `laplace` package (Daxberger et al., 2021a), an easy-to-use software library for PyTorch offering access to the most common LA methods, discussed in Section 4.3. The server performs 300 steps to minimize the global loss using the Adam optimizer with its standard default hyperparameters. During each server's optimizer run, conducted separately for each ensemble model, the validation performance is evaluated every 30 steps and the parameters configuration that achieves the best validation performance is selected as the final component of the ensemble. Last, for all the experiments we employed a (cold) tempered posterior for each client, with $T = 0.1$, and a diagonal Gaussian as prior with variance $\sigma^2 = 0.1$. An ablation study on these two hyperparameters can be found in the Appendix.

All the experiments were performed on a machine equipped with an Intel Xeon 4114 CPU and NVIDIA Titan XP GPU with 12Gb of RAM. Our code is available at: `https://github.com/jacopot96/FedBEns`

# 6. Numerical Results

## 6.1. Varying heterogeneity

In this Section, we compare the different methods in a scenario where the number of clients is fixed ($C = 5$) while varying the heterogeneity parameter $\alpha$. We used the same

*Table 1.* Test set performance metrics, 5 clients, varying heterogeneity parameter

| Dataset | FedBEns $M = 5$ Kron | FedBEns $M = 5$ Diag+Full | FedFisher K-FAC | RegMean | DENSE | OTFusion | Fisher Merge |
|---|---|---|---|---|---|---|---|
| | | Mean Accuracy & Standard Deviation (5 seeds) | | | | | |
| | | | $\alpha = 0.05$ | | | | |
| FMNIST | $\mathbf{59.89 \pm 3.61}$ | $\underline{55.58 \pm 2.96}$ | $53.92 \pm 1.89$ | $48.04 \pm 4.18$ | $38.48 \pm 5.56$ | $37.26 \pm 1.01$ | $43.11 \pm 6.18$ |
| SVHN | $\mathbf{71.55 \pm 1.65}$ | $53.97 \pm 1.87$ | $58.46 \pm 3.70$ | $\underline{65.57 \pm 1.83}$ | $56.89 \pm 0.75$ | $39.17 \pm 0.42$ | $43.53 \pm 2.54$ |
| CIFAR10 | $\mathbf{51.43 \pm 0.79}$ | $33.69 \pm 3.38$ | $\underline{36.97 \pm 2.01}$ | $33.77 \pm 0.52$ | $30.61 \pm 1.95$ | $29.59 \pm 1.77$ | $28.44 \pm 3.37$ |
| Average | $\mathbf{60.90}$ | $47.75$ | $\underline{49.58}$ | $49.13$ | $41.99$ | $35.33$ | $38.36$ |
| | | | $\alpha = 0.1$ | | | | |
| FMNIST | $\mathbf{75.91 \pm 3.21}$ | $\underline{70.04 \pm 1.13}$ | $68.06 \pm 2.64$ | $56.40 \pm 5.54$ | $47.16 \pm 1.73$ | $43.54 \pm 2.43$ | $57.38 \pm 5.66$ |
| SVHN | $\mathbf{80.67 \pm 0.48}$ | $63.97 \pm 1.51$ | $65.52 \pm 1.42$ | $\underline{71.01 \pm 1.29}$ | $62.65 \pm 0.91$ | $51.94 \pm 0.67$ | $52,66 \pm 4.04$ |
| CIFAR10 | $\mathbf{57.62 \pm 1.24}$ | $46.68 \pm 2.64$ | $\underline{46.92 \pm 1.79}$ | $40.86 \pm 0.77$ | $38.04 \pm 1.78$ | $40.41 \pm 0.73$ | $36.56 \pm 1.51$ |
| Average | $\mathbf{71.40}$ | $\underline{60.23}$ | $60.17$ | $58.13$ | $49.32$ | $45.30$ | $48.87$ |
| | | | $\alpha = 0.2$ | | | | |
| FMNIST | $\mathbf{79.03 \pm 2.71}$ | $\underline{77.24 \pm 1.88}$ | $72.89 \pm 3.19$ | $71.43 \pm 1.61$ | $69.07 \pm 4.23$ | $57.06 \pm 2.19$ | $66.89 \pm 3.50$ |
| SVHN | $\mathbf{82.91 \pm 0.26}$ | $77.71 \pm 0.31$ | $71.61 \pm 5.14$ | $\underline{78.24 \pm 0.81}$ | $77.13 \pm 3.39$ | $72.19 \pm 1.00$ | $68.72 \pm 0.68$ |
| CIFAR10 | $\mathbf{59.78 \pm 0.92}$ | $49.12 \pm 1.15$ | $\underline{49.95 \pm 2.51}$ | $43.42 \pm 1.54$ | $44.94 \pm 2.50$ | $40.64 \pm 2.07$ | $40.54 \pm 3.11$ |
| Average | $\mathbf{73.91}$ | $\underline{68.02}$ | $63.72$ | $64.36$ | $64.81$ | $56.63$ | $58.72$ |
| | | | $\alpha = 0.4$ | | | | |
| FMNIST | $\mathbf{84.42 \pm 0.77}$ | $\underline{81.81 \pm 0.26}$ | $74.76 \pm 2.33$ | $74.45 \pm 1.89$ | $77.91 \pm 1.87$ | $67.29 \pm 4.78$ | $78.16 \pm 0.62$ |
| SVHN | $\mathbf{85.58 \pm 0.15}$ | $\underline{82.15 \pm 0.71}$ | $78.31 \pm 0.32$ | $80.11 \pm 1.03$ | $78.40 \pm 2.49$ | $73.79 \pm 3.25$ | $73.34 \pm 1.77$ |
| CIFAR10 | $\mathbf{61.49 \pm 0.72}$ | $48.96 \pm 0.16$ | $\underline{51.89 \pm 1.99}$ | $41.82 \pm 3.70$ | $46.47 \pm 2.07$ | $45.84 \pm 0.30$ | $41.78 \pm 3.98$ |
| Average | $\mathbf{77.16}$ | $\underline{70.92}$ | $68.32$ | $65.46$ | $67.59$ | $62.31$ | $64.43$ |

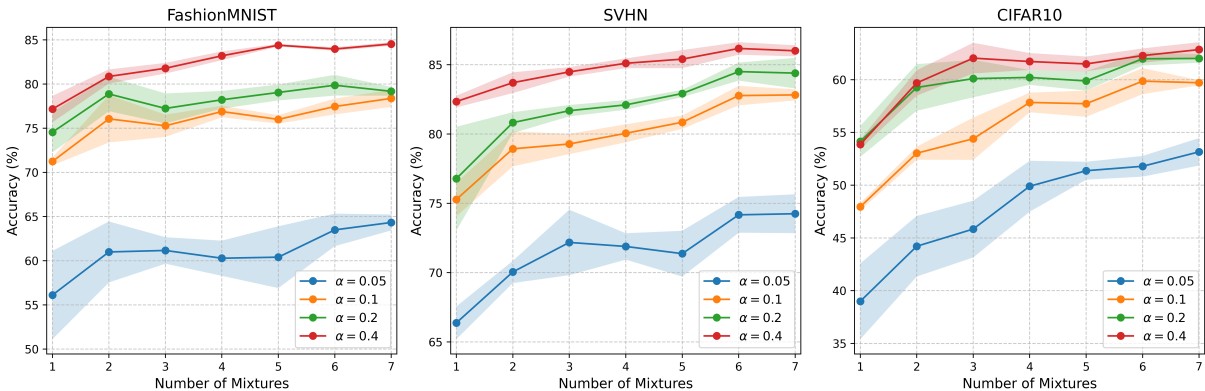

*Figure 2.* Test accuracy as a function of the number of mixtures for FedBEns with Kronecker factorization. For each dataset, the mean accuracy is represented by a solid line, while the shaded band indicates the standard deviation, both computed over 5 seeds, for various heterogeneity parameters.

parameters across all settings, such as the number of local epochs, to ensure a fair comparison. Specifically, we considered settings ranging from high to moderate heterogeneity ($\alpha = \{0.005, 0.1, 0.2, 0.4\}$) (Jhunjhunwala et al., 2024) to analyze their impact on the different approaches.

Table 1 shows the average accuracy along with the standard deviation for our method with a fixed number of mixtures, $M = 5$, and compare it with the other considered baselines. Results were obtained over 5 experiments with different random seeds. Additionally, we report the average accuracy across datasets for each model and experimental setting to facilitate a clearer comparison. Bold numbers indicate the highest accuracy, while underlined numbers represent the

*Table 2.* Test set performance metrics, $\alpha = 0.3$, varying number of clients

| Dataset | FedBEns $M = 5$ Kron | FedBEns $M = 5$ Diag+Full | FedFisher K-FAC | RegMean | DENSE | OTFusion | Fisher Merge |
|---|---|---|---|---|---|---|---|
| | Mean Accuracy & Standard Deviation (5 seeds) | | | | | | |
| | $C = 10$ | | | | | | |
| FMNIST | **80.57 ± 0.86** | 78.66 ± 1.73 | 73.85 ± 3.91 | 73.97 ± 3.18 | 74.33 ± 3.95 | 64.34 ± 4.54 | 66.84 ± 6.07 |
| SVHN | **83.75 ± 0.22** | 75.98 ± 1.85 | 74.94 ± 0.69 | 78.64 ± 0.95 | 64.99 ± 5.75 | 67.76 ± 3.39 | 63.99 ± 0.39 |
| CIFAR10 | **58.14 ± 1.19** | 48.18 ± 1.52 | 52.14 ± 1.73 | 44.48 ± 1.74 | 45.39 ± 3.18 | 41.39 ± 1.28 | 38.10 ± 2.58 |
| Average | **74.15** | 67.61 | 66.98 | 65.65 | 61.57 | 57.83 | 56.31 |
| | $C = 20$ | | | | | | |
| FMNIST | **80.83 ± 0.54** | 76.16 ± 1.04 | 72.83 ± 3.21 | 72.91 ± 1.56 | 74.26 ± 3.89 | 63.46 ± 6.44 | 64.11 ± 4.49 |
| SVHN | **81.21 ± 0.85** | 74.33 ± 3.72 | 78.27 ± 0.48 | 79.04 ± 0.89 | 58.93 ± 3.81 | 59.25 ± 4.66 | 62.23 ± 3.18 |
| CIFAR10 | **56.41 ± 1.81** | 43.39 ± 2.58 | 48.85 ± 0.84 | 42.81 ± 2.37 | 43.75 ± 1.91 | 38.98 ± 1.69 | 33.69 ± 2.22 |
| Average | **72.77** | 64.63 | 66.65 | 64.79 | 58.98 | 53.90 | 53.34 |
| | $C = 40$ | | | | | | |
| FMNIST | **77.27 ± 0.84** | 72.69 ± 1.55 | 72.06 ± 2.73 | 72.09 ± 1.05 | 71.81 ± 1.23 | 64.48 ± 4.27 | 64.29 ± 4.33 |
| SVHN | **80.06 ± 0.56** | 65.96 ± 2.97 | 76.85 ± 0.98 | 76.94 ± 0.86 | 59.84 ± 3.76 | 50.91 ± 4.48 | 52.56 ± 5.18 |
| CIFAR10 | **54.87 ± 1.72** | 42.38 ± 1.45 | 47.76 ± 0.69 | 44.35 ± 0.97 | 41.35 ± 1.83 | 34.53 ± 3.02 | 33.81 ± 2.56 |
| Average | **70.73** | 59.68 | 65.56 | 64.46 | 57.67 | 49.97 | 50.22 |

second-best performance. From this table, it is possible to conclude that FedBEns with Kronecker factorization, systematically outperforms the baselines. This highlights the effectiveness of FedBEns as a One-Shot FL algorithm, especially in scenarios with high data heterogeneity.

To further examine the impact of the number of mixture components $M$, Figure 2 shows the accuracy of FedBEns with Kronecker factorization of the Hessian as a function of the ensemble cardinality. The results demonstrate an almost monotonic improvement in accuracy as the number of components increases until a plateau is reached, with a significant boost observed even with just two or three components. Notably, our proposed approach achieves competitive performance with respect to state-of-the-art methods while only using a few components, and often surpasses them with just two components. In fact, the accuracy of FedBEns with a single mixture ($M = 1$) is comparable to, or slightly better than, the baselines across all datasets and heterogeneity levels (compare Figure 2 and Table 1), and FedBEns outperforms the baselines already for $M = 2$.

We also conducted an ablation study on Hessian approximations, detailed in the Appendix, highlighting the advantages of Kronecker factorization over other methods.

The possibility of choosing the number of mixtures adds a degree of flexibility to control the trade-off between computational/communication costs (higher when the number of components is higher) and accuracy, making our approach suitable to real-world applications with widely different requirements. Further analyses and discussions of

this computational-accuracy tradeoff are provided in the Appendix. Additionally, we compare our approach with FedFisher when run over multiple communication rounds, ensuring comparable total communication costs between the two approaches. We also conduct a separate comparison where we equalize client computation costs by running FedBEns on smaller datasets. Even in these settings, FedBEns demonstrates competitive performance (see the Appendix for more details).

### 6.2. Varying number of clients

Table 2 reports the results for varying the number of clients ($C \in \{10, 20, 40\}$), with the heterogeneity parameter fixed at $\alpha = 0.3$, to provide insights into the behavior of the proposed approach as the number of clients increases.

It is worth noting that, in this scenario, each client dataset is smaller, which increases the risk of local models overfitting and makes the aggregation more challenging. From this analysis emerges that FedBEns with Kronecker factorization outperforms the other baselines.

Overall, our approach demonstrates highly competitive performance, outperforming state-of-the-art models, by up to 10 percentage points in some cases, across different heterogeneity settings and client numbers.

## 7. Conclusion and Future Work

In this paper, we proposed a novel approach for One-Shot Federated Learning based on Bayesian inference. Our

method utilizes a mixture of Laplace approximations to model client posteriors, which are used to estimate the global posterior. Our proposed approach outperforms other state-of-the-art models, with respect to classification accuracy, across different scenarios.

Many interesting research directions remain open for exploration. One of the main limitations of the proposed approach is its computational cost, since it is based on a mixture of $M$ models trained in parallel instead of just one. However, it is worth mentioning that there are approaches that mitigate this problem (e.g. (Havasi et al., 2021)) by mimicking ensemble prediction within a single model. Moreover, we plan to study the privacy risks associated with the proposed approach and try to enhance it, e.g., through differential privacy techniques. Additionally, we would like to study the adaptability of FedBEns to dynamic environments where, for instance, new clients join the Federated Learning scheme and/or clients collect new types of data and experience a distribution shift.

## Acknowledgement

The research leading to these results has been partially funded by the Italian Ministry of University and Research (MUR) under the PRIN 2022 PNRR framework (EU Contribution – NextGenerationEU – M. 4,C. 2, I. 1.1), SHIELDED project, ID P2022ZWS82; by the French government, through the 3IA Côte d'Azur Investments in the Future project managed by the National Research Agency (ANR) with the reference number ANR-19-P3IA-0002; by the European Network of Excellence dAIEDGE under Grant Agreement Nr. 101120726; by the EU HORIZON MSCA 2023 DN project FINALITY (G.A. 101168816); by the Groupe La Poste, sponsor of the Inria Foundation, in the framework of the FedMalin Inria Challenge.

## Impact Statement

This work advances the field of machine learning by introducing an algorithm that enables Federated Learning with a single round of communication between a central server and clients. We believe this work mainly aligns with the increasing demand for user privacy in widely different technological domains, which is one of the main advantages of the Federated Learning paradigm. This has many potential societal consequences of our work, none of which we feel must be specifically highlighted here.

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

# A. Proof: Global Posterior

**Proposition A.1.** *Given $C$ datasets and the posteriors $p(\mathbf{w}|\mathcal{D}_c)$ of the same model $f(\mathbf{w})$, under the assumptions that (i) the datasets are conditionally independent given the model, and (ii) the prior is the same across clients, the global posterior can be written as:*

$$p(\mathbf{w}|\mathcal{D}_1, .., \mathcal{D}_C) = \alpha \cdot \frac{\prod_{c=1}^{C} p(\mathbf{w}|\mathcal{D}_c)}{p(\mathbf{w})^{C-1}} \tag{7}$$

**Proof:**

The starting point is the global likelihood function $p(\mathcal{D}|\mathbf{w})$. Under the assumption that clients are *conditionally* independent, it can be decomposed as:

$$p(\mathcal{D}|\mathbf{w}) \equiv p(\mathcal{D}_1, .., \mathcal{D}_C|\mathbf{w}) = \prod_{c=1}^{C} p(\mathcal{D}_c|\mathbf{w}) \tag{8}$$

To obtain the posterior, it is possible to rely on Bayes' theorem $p(\mathbf{w}|\mathcal{D}) = \frac{p(\mathcal{D}|\mathbf{w})p(\mathbf{w})}{p(\mathcal{D})}$ :

$$p(\mathbf{w}|\mathcal{D}) = p(\mathbf{w}|\mathcal{D}_1, .., \mathcal{D}_C) = \frac{\prod_{c=1}^{C} p(\mathcal{D}_c|\mathbf{w}) \cdot p(\mathbf{w})}{p(\mathcal{D}_1, .., \mathcal{D}_C)} \tag{9}$$

Equivalently, instead of combining the likelihoods, it is possible to rewrite the previous Equation in terms of each client posterior, exploiting Bayes' theorem again:

$$p(\mathbf{w}|\mathcal{D}_1, .., \mathcal{D}_C) = \frac{p(\mathbf{w})}{p(\mathcal{D}_1, .., \mathcal{D}_C)} \prod_{c=1}^{C} p(\mathcal{D}_c|\mathbf{w}) \tag{10}$$

$$= \frac{p(\mathbf{w})}{p(\mathcal{D}_1, .., \mathcal{D}_C)} \prod_{c=1}^{C} \frac{p(\mathcal{D}_c)p(\mathbf{w}|\mathcal{D}_c)}{p(\mathbf{w})} \tag{11}$$

$$= \underbrace{\frac{\prod_{c=1}^{C} p(\mathcal{D}_c)}{p(\mathcal{D}_1, .., \mathcal{D}_C)}}_{\equiv \alpha} \frac{\prod_{c=1}^{C} p(\mathbf{w}|\mathcal{D}_c)}{p(\mathbf{w})^{C-1}} \tag{12}$$

$\square$

**Remark 1.** This expression can be generalized to different local priors, denoted as $p_c(\mathbf{w})$, as:

$$p(\mathbf{w}|\mathcal{D}_1, .., \mathcal{D}_C) = \alpha \cdot p_{\text{global}}(\mathbf{w}) \cdot \prod_{c=1}^{C} \frac{p(\mathbf{w}|\mathcal{D}_c)}{p_c(\mathbf{w})} \tag{13}$$

where $p_{\text{global}}(\mathbf{w})$ represent the global prior, provided by the server.

**Remark 2.** The conditional independence assumption plays a crucial role. Otherwise, one has to model the joint likelihood explicitly, since $p(\mathcal{D}_1, .., \mathcal{D}_C|\mathbf{w}) \neq \prod_{c=1}^{C} p(\mathcal{D}_c|\mathbf{w})$. A possible approach is to consider an autoregressive-like likelihood. For instance, for two clients, a correlated log-likelihood can be derived from a Gaussian-like approximation of joint likelihood, as:

$$\log(p(\mathcal{D}_1, \mathcal{D}_2|\mathbf{w})) = \log(p(\mathcal{D}_1|\mathbf{w})) + \log(p(\mathcal{D}_2|\mathbf{w})) + \rho \cdot g(\mathcal{D}_1, \mathcal{D}_2, \mathbf{w}) \tag{14}$$

where: $\rho \in [-1, 1]$ is a correlation coefficient and $g$ is a function that captures the functional dependencies among clients' data.

**Remark 3.** The global posterior in Equation (2) factorizes independently across clients. Consequently, incorporating new clients into the federation or forgetting clients that leave the federation is straightforward: it is sufficient to simply add or remove their corresponding mixture component from the global posterior.

## B. Additional Related Work

In this section, we discuss additional related work on FL, with a focus on papers based on an ensemble of models. An early contribution that leverages ensemble and Bayesian inference in the standard multi-rounds FL setting is FedBE (Chen & Chao, 2021), where the aggregation at the server is based on the local posterior approximated as a Gaussian or a Dirichlet distribution, relying on the stochastic weight average-Gaussian (SWAG) method (Maddox et al., 2019). Models are then sampled from the global (unimodal) distribution and the knowledge of this ensemble is then distilled in a single model, under the assumption that a certain amount of unlabelled data is available at the server.

Regarding other approaches that exploit ensembles in a Federated Learning scenario, it is worth mentioning FENS (Allouah et al., 2024), FedKD (Gong et al., 2022), Fed-ET (Cho et al., 2022), FedBoost (Hamer et al., 2020), and FedeDF (Lin et al., 2020). FENS is a One-Shot FL method that operates in two stages: first, clients train models locally and send them to the server; second, clients collaboratively train a lightweight prediction aggregator model using standard multi-rounds FL. Other approaches (e.g, FedKD and Fed-ET) are designed for standard multi-rounds FL settings where knowledge distillation is exploited to reduce communications costs (Gong et al., 2022) and/or improve generalization by introducing a data-aware weighted consensus from the ensemble of models, with a feedback loop to transfer the server model's knowledge to the client models (Cho et al., 2022).

Our approach differs from these works as it does not require data on the server side or additional server-clients communication rounds. Moreover, the underlying theoretical motivation is also different: our method leverages ensembles to achieve a tractable and effective approximation of the posterior and its Bayesian marginalization, whereas other approaches employ ensembles guided principles of knowledge distillation and/or stacked generalization.

## C. Ablation and Further Studies

### C.1. Hessian matrix approximations

One key design feature of the proposed approach is the number of mixtures, $M$. Increasing the ensemble size improves predictive performance but raises server-client communication costs. Additionally, also the choice of the approximation structure for the Hessian matrix estimation impacts performance and computation. Here we consider the following approximations:

- **Diagonal**. It is the simplest and most scalable approach that ignores off-diagonal terms of the Hessian. Denoting the total number of parameters of the model as $d$, diagonal approximation requires additional $d$ parameters for the Hessian matrix.

- **Diagonal+Full**. It adds expressivity to the previous diagonal approximation by considering a full Hessian for the last layer. In this case, the computational cost raises w.r.t the diagonal approximation by an amount that scales with the last-layer number of neurons, $n_{\text{last}}$, and the number of classes, $n_{\text{classes}}$, as $\mathcal{O}(n_{\text{last}} \cdot n_{\text{classes}})^2$

- **Kronecker.** As we discussed in Section 4.3, it is one of the most common approximations of the Hessian since it is a good trade-off between computational cost and expressivity (Daxberger et al., 2021a).

In general, the diagonal approximation is less expensive, while the comparison between Kronecker and Diagonal+last Full depends on the model architecture. For instance, for the considered CNN employed on SVHN and CIFAR10 and denoting, as said, the total number of parameters as $d$, Kronecker factorization requires $\sim 3.9d$ parameters, while Diagonal+last Full approximation $\sim 2.7d$. However, for other architectures (e.g. those with a substantially higher number of classes), Kronecker may become more cost-effective.

To assess the impact of each approximation on predictive performance, we conducted an ablation study examining three types of approximations across different numbers of mixtures. Our analysis focused on two different heterogeneity scenarios, i.e., $\alpha = 0.05$ (highly heterogeneous), and $\alpha = 0.4$ (medium/low heterogeneity). Figure 3 shows the results. As expected,

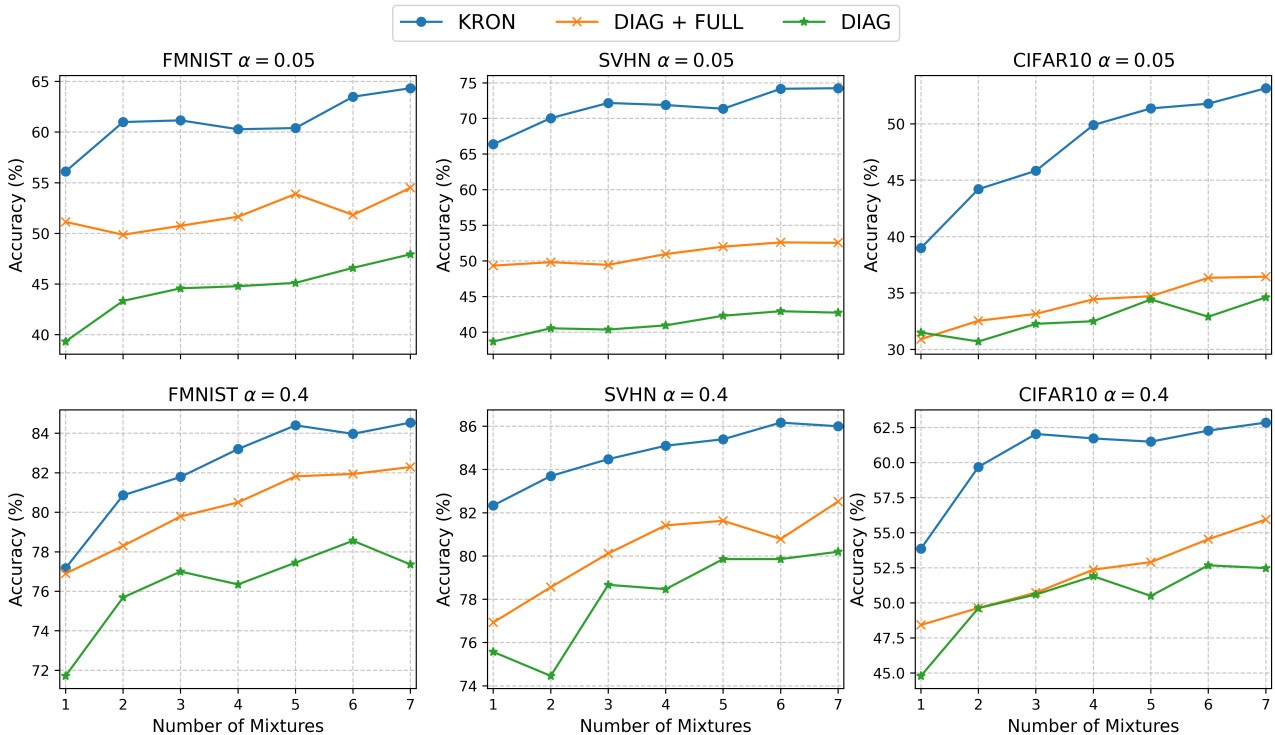

*Figure 3.* Test accuracy as a function of the number of mixtures with different Hessian approximations. Curves represent the mean accuracy computed over 3 seeds.

the diagonal approximation yields the worst performance, while the Kronecker factorization consistently achieves the highest accuracy, highlighting the importance of capturing correlations among model parameters for each layer.

### C.2. Temperature parameter

In Section 4.3 we introduced temperature scaling and, particularly, we asserted that cold temperature $T < 1$ ensures to obtain a more concentrated posterior that prevents Laplace approximation from giving too much mass to certain regions (e.g. due to the approximation of the Hessian). Figure 4 presents the results of an ablation study using our approach with 5 mixtures and Kronecker Hessian approximations. In general, high temperatures (i.e., $T > 1$) result in poor performance, while low temperatures tend to yield better results, also demonstrating a certain robustness with respect to the temperature choice. It is worth noting that, in principle, the performance of our approach could be further improved by tuning the temperature parameter for each experiment (e.g. through a grid search on the server if a validation dataset is available), while the results we presented are with a fixed temperature of 0.1.

### C.3. Prior variance

We present an ablation study on the variance of the isotropic Gaussian prior $p(\mathbf{w}) = \mathcal{N}(0, \sigma^2 I)$. Figure 5 reports the results for FedBEns with Kronecker factorization, 5 mixtures, and $T = 0.1$ on 5 clients. The predictive performance remains robust across a reasonably wide range of prior variance values.

### C.4. On the role of the validation set

As stated in Section 5.3, our method uses a small validation set during the server's gradient descent runs, to find the parameter configuration that achieves the best validation performance. In the following, we report an ablation study where no validation dataset is used and the number of server epochs is fixed to 100. The results are reported in Table 3. Remarkably, FedBEns still outperforms the considered baselines, which continue to leverage the validation dataset on the server side to

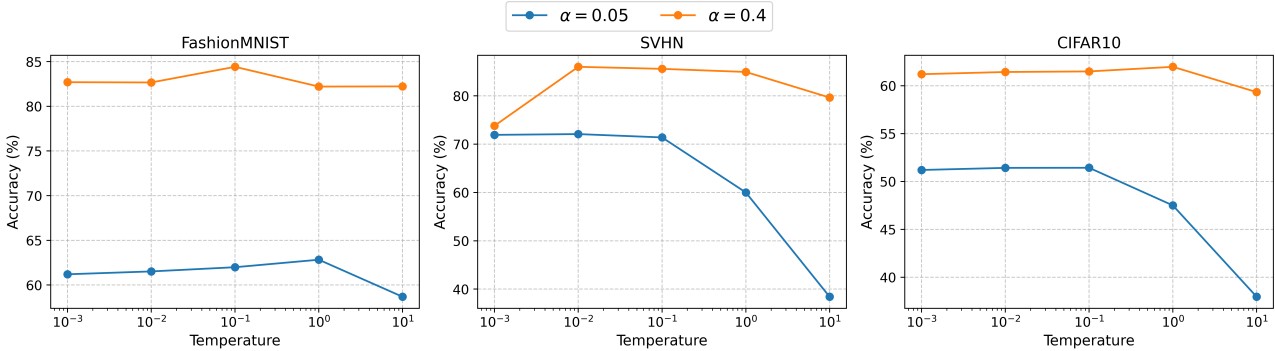

*Figure 4.* Test accuracy for FedBEns with Kronecker factorization and 5 mixtures, as a function of the temperature parameter. Curves represent the mean accuracy computed over 3 seeds.

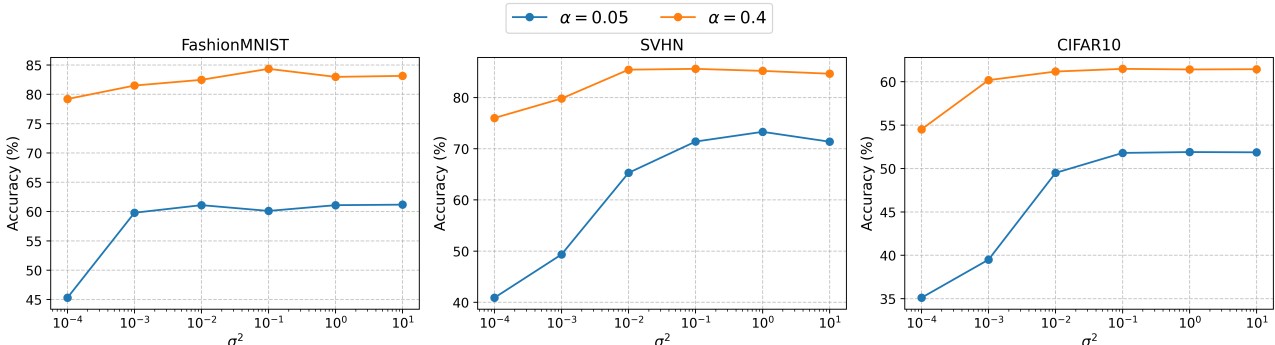

*Figure 5.* Test accuracy for FedBEns with Kronecker factorization and 5 mixtures, as a function of the prior variance $\sigma^2$. Curves represent the mean accuracy computed over 3 seeds.

tune certain hyperparameters depending on the approach, with only a minor drop in accuracy, up to a few p.p. (see Table 1 in the paper and Table 3 below). This makes FedBEns applicable even in scenarios where such a small amount of data is not available at server side.

### C.5. Computational and communication costs

In this Section, we comment on the computational and communication costs associated with the different approaches. We observe that the FedBEns per-client computational and communication costs are both linear in the number of mixtures $M$. On the contrary, the computation time at the server scales as $M^2$, since there is the need to find $M$ modes of the global log-posterior through $M$ SGD runs, and the cost of each evaluation of the global log-posterior scales linearly with $M$ (see Equation (4)).

In Table 4 we provide the communication cost and a wall-clock time comparison with the other baselines computed on CIFAR10 with 5 clients on the hardware specified in Sec. 5.3. The reported values are the average over 5 runs. Overall, our approach increases computation and communication costs depending on the number of mixture components, but it also improves predictive performance. Whether this additional cost is justified depends on the practical context. In the following paragraphs, we provide further insights into the trade-off between predictive performance and computational cost.

*Table 3.* Test set performance metrics of FedBEns with no validation set

| Dataset | FedBEns $M = 5$ Kron |
|---|---|
| Mean Accuracy & Standard Deviation (3 seeds) | |
| $\alpha = 0.05$ | |
| FMNIST | $59.15 \pm 3.87$ |
| SVHN | $70.26 \pm 1.71$ |
| CIFAR10 | $48.81 \pm 0.98$ |
| $\alpha = 0.4$ | |
| FMNIST | $82.16 \pm 0.83$ |
| SVHN | $84.69 \pm 0.27$ |
| CIFAR10 | $60.00 \pm 0.75$ |

*Table 4.* Computational/Communication costs

| Method | Client runtime (s) | Server runtime (s) | Communication cost (MB) |
|---|---|---|---|
| FedBEns Kron | $30.9 \cdot M$ | $7.2 \cdot M^2$ | $17.8 \cdot M$ |
| FedFisher K-FAC | 30.9 | 13.4 | 17.8 |
| RegMean | 26.0 | 5.1 | 7.3 |
| DENSE | 25.9 | 170.2 | 3.65 |
| OTFusion | 25.9 | 1.1 | 3.65 |
| FisherMerge | 26.2 | 0.9 | 7.31 |

## C.6. Mixture in One-Shot vs. multiple rounds

In this Section, we compare the proposed approach with its main competitor, FedFisher, while ensuring the same server-client communication cost. In fact, FedFisher has been extended to support multiple rounds of server-client communication, as detailed in (Jhunjhunwala et al., 2024). Specifically, we compare FedFisher K-FAC with 5 rounds of communication to our approach using 5 mixtures in the One-Shot setting. In Table 5 we report the accuracy for the described methods in two scenarios, i.e., high ($\alpha = 0.05$) and medium/low ($\alpha = 0.4$) heterogeneity.

Although the fairness of this comparison (one round of communication vs. multiple rounds) may be questionable, it is remarkable that FedBEns gives competitive results, especially in a more balanced scenario. Note that even though the communication cost is the same, sending all the needed information in a single round of communication, if the client-server communication channel has sufficient capacity, may be preferable in specific cases, e.g. when the client-server connection is not continuously available or possible. Moreover, while FedBEns with $M$ mixtures and FedFisher with $M$ rounds entail similar per-client communication/computation costs, the total time a client must remain active during training can differ significantly due to server-to-client transmissions and the straggler effect.

First, consider server-to-client communication. Assuming equal server/clients uplink and downlink capacities, the communication time for $M$-round FedFisher scales as $2M$, since the model must be sent from client to server and back in each round. In contrast, FedBEns communication time is halved since mixtures are sent once and then aggregated. If communication is the system bottleneck (as is often the case in FL), training with $M$-round FedFisher would require each client to remain available for roughly twice as long as FedBEns with $M$ mixtures. Notably, the final model in FedBEns remains unchanged even if a client disconnects after submitting its mixtures.

Second, consider the straggler effect. The reasoning above assumes homogeneous clients and synchronization. However, real-world systems are affected by stragglers, clients whose updates arrive significantly later than those of others (Kairouz et al., 2021). Stragglers slow down the entire system, as the server must wait for their inputs before generating a new model to distribute. This delay, in turn, forces other clients to remain idle, further extending their total participation time. One-shot algorithms mitigate this issue, as clients can disconnect immediately after submitting their contributions.

Finally, It is also important to highlight the value of minimizing client participation time. In a cross-device setting, clients often exhibit volatile participation patterns. Therefore, minimizing the required client's participation time is critical: it may

*Table 5.* Test set performance metrics, multiple mixtures vs. multiple rounds

| Dataset | FedBEns $M = 5$ Kron | FedFisher K-FAC 5 communication rounds |
|---|---|---|
| Mean Accuracy & Standard Deviation (3 seeds) | | |
| $\alpha = 0.05$ | | |
| FMNIST | $60.08 \pm 3.60$ | $\mathbf{66.49 \pm 4.89}$ |
| SVHN | $\mathbf{71.39 \pm 1.63}$ | $67.27 \pm 2.05$ |
| CIFAR10 | $51.38 \pm 0.79$ | $\mathbf{58.28 \pm 1.03}$ |
| $\alpha = 0.4$ | | |
| FMNIST | $\mathbf{84.39 \pm 0.78}$ | $82.3 \pm 0.94$ |
| SVHN | $\mathbf{85.57 \pm 0.15}$ | $83.06 \pm 1.53$ |
| CIFAR10 | $61.56 \pm 0.73$ | $\mathbf{68.14 \pm 2.03}$ |

*Table 6.* Test set performance metrics, multiple mixtures on smaller datasets

| Dataset | FedBEns $M = 5$ Kron Reduced Datasets | FedFisher K-FAC |
|---|---|---|
| Mean Accuracy & Standard Deviation (3 seeds) | | |
| $\alpha = 0.05$ | | |
| FMNIST | $50.87 \pm 4.46$ | $\mathbf{53.92 \pm 1.89}$ |
| SVHN | $\mathbf{59.12 \pm 1.42}$ | $58.46 \pm 3.70$ |
| CIFAR10 | $\mathbf{45.38 \pm 0.62}$ | $36.97 \pm 2.01$ |
| $\alpha = 0.4$ | | |
| FMNIST | $\mathbf{77.95 \pm 0.31}$ | $74.76 \pm 2.33$ |
| SVHN | $\mathbf{81.15 \pm 0.25}$ | $78.31 \pm 0.32$ |
| CIFAR10 | $\mathbf{53.82 \pm 1.16}$ | $51.89 \pm 1.99$ |

determine whether a client contributes to the training or drops out before completion, thereby preventing the federation from leveraging its dataset and degrading the final model's quality (Kairouz et al., 2021). Finally, we observe that in FedBEns each client can transmit each mixture as soon as it is computed. As a result, even if a client disconnects before completing all $M$ mixtures, the system may benefits from its partial contribution (see also C.8).

### C.7. Mixture in One-Shot on smaller datasets vs. single model on larger datasets

In this Section, we present an additional comparison with FedFisher, where we equalize the client computation of our approach by running FedBEns with 5 mixture components on client training datasets that are 5 times smaller. Remarkably, FedBEns performs better than FedFisher in most scenarios, as illustrated by the results reported in Table 6.

### C.8. System heterogeneity: customizing the number of mixtures for each client

As previously discussed, FedBEns introduces a computational overhead that depends on the number of mixture components. In practical scenarios, clients may have different computational capabilities, so this additional computation could be an issue for some of them, but not for others. In this Section, we explore the possibility of customizing the number of mixtures based on each client's computational power. As an illustrative example, we considered a setting with 5 clients: 3 can only employ a single Laplace approximation ($M = 1$) and the remaining 2 are able to exploit three mixtures ($M = 3$). We considered two different approaches. In the first, all 5 clients use a unimodal approximation of the posterior ($M = 1$). In the second, the number of mixture components is customized: 3 clients use $M = 1$, while the remaining 2 use $M = 3$, obtained by running 3 SGD optimizer in parallel from the same initialization. In both cases, the final model returned to each client consists of a single model.

Table 7 reports the results in the described settings, indicating that the additional information coming from the clients with

*Table 7.* Test set performance metrics, customizing the number of mixtures

| Dataset | FedBEns Kron $M = 1$ | FedBEns Kron $M$ customized |
|---|---|---|
| | Mean Accuracy & Standard Deviation (3 seeds) | |
| | $\alpha = 0.05$ | |
| FMNIST | $57.02 \pm 3.73$ | $59.57 \pm 1.44$ |
| SVHN | $66.11 \pm 0.77$ | $68.39 \pm 1.57$ |
| CIFAR10 | $39.89 \pm 1.71$ | $42.32 \pm 2.25$ |
| | $\alpha = 0.4$ | |
| FMNIST | $77.07 \pm 0.29$ | $78.31 \pm 1.76$ |
| SVHN | $82.86 \pm 0.21$ | $82.75 \pm 1.41$ |
| CIFAR10 | $53.97 \pm 0.86$ | $54.68 \pm 1.88$ |

*Table 8.* Test set performance metrics, ensemble vs. single model

| Dataset | Ensemble | Min | Max |
|---|---|---|---|
| | Mean Accuracy & Standard Deviation (3 seeds) | | |
| | $\alpha = 0.05$ | | |
| FMNIST | $60.08 \pm 3.60$ | $53.41 \pm 1.78$ | $59.98 \pm 1.68$ |
| SVHN | $71.39 \pm 1.63$ | $62.24 \pm 1.67$ | $68.99 \pm 1.4$ |
| CIFAR10 | $51.38 \pm 0.79$ | $36.21 \pm 4.68$ | $42.77 \pm 0.32$ |
| | $\alpha = 0.4$ | | |
| FMNIST | $84.39 \pm 0.78$ | $76.69 \pm 0.95$ | $79.87 \pm 0.76$ |
| SVHN | $85.57 \pm 0.15$ | $81.25 \pm 0.95$ | $83.38 \pm 0.11$ |
| CIFAR10 | $61.56 \pm 0.73$ | $49.22 \pm 1.60$ | $54.27 \pm 0.81$ |

more computational power benefits the entire federation. This advantage allows the server to find a better global model, especially in heterogeneous settings.

### C.9. Ensemble vs. single component for global model

As discussed in Section 3, our approach leverages a multimodal approximation of the global posterior and exploits an ensemble of final models, in line with a Bayesian Model Averaging perspective. In this Section, we analyze how much the ensemble contributes to enhance the predictive performance compared to selecting, for example, the ensemble component with the highest accuracy.

In Table 8 we present the final accuracy of the ensemble, along with the highest (Max) and lowest (Min) accuracies achieved by its individual components. These results highlight the benefits of ensembling, as relying on the solution provided by a single SGD run on the global loss may result in suboptimal performance.

### C.10. A more challenging setting: ResNet18 on CIFAR100

We also conducted an experiment on a more challenging task, represented by CIFAR-100 (Krizhevsky et al., 2010). As a base model, we used the ResNet18 architecture (He et al., 2016) without batch normalization to ensure compatibility with the different One-Shot approaches. Training was conducted for 100 epochs for each client. Given our hardware limitations, we were able to run FedBEns with only 2 mixtures using Kronecker Factorization. Table 9 summarizes the performance of 5 clients in two different heterogeneity settings. As a reference, the employed model achieves an accuracy of $58.55\%$ in the usual centralized setting. Even in this more complex task, in which training in One-Shot is challenging, FedBEns shows the highest performance, followed by the DENSE algorithm. In particular, it is worth mentioning that at the cost of doubling the computation/communications cost w.r.t. FedFisher, FedBEns achieves a higher gain in accuracy (doubled or even more).

*Table 9.* Test set performance metrics, 5 clients, varying heterogeneity parameter

| Mean Accuracy & Standard Deviation (3 seeds) | | | | | | |
|---|---|---|---|---|---|---|
| Dataset | FedBEns $M = 2$ Kron | FedFisher K-FAC | RegMean | DENSE | OTFusion | Fisher Merge |
| $\alpha = 0.05$ | | | | | | |
| CIFAR100 | **8.56 ± 0.47** | 3.14 ± .23 | 3.01 ± 0.54 | 5.88 ± 1.07 | 4.79 ± 0.19 | 2.81 ± 0.33 |
| $\alpha = 0.4$ | | | | | | |
| CIFAR100 | **17.85 ± 0.67** | 4.79 ± 0.31 | 4.85 ± 1.16 | 13.92 ± 0.54 | 4.74 ± 0.56 | 3.87 ± 0.48 |
| Communication costs (MB) | 716 | 358 | 93 | 47 | 47 | 94 |
| Local runtime (min) | 28.0 | 14.0 | 9.3 | 9.2 | 9.2 | 10.1 |
| Server runtime (min) | 2.3 | 1.2 | 0.6 | 10.4 | 0.2 | 0.3 |

