# OpenReview forum: "FedBEns: One-Shot Federated Learning based on Bayesian Ensemble"
_ICML.cc/2025/Conference — ICML 2025 poster_

### Official Review · Reviewer_dxFK · 2025-03-09

**Overall Recommendation:** 3

**Summary:**

FedBEns proposes a one-shot federated learning utilizing a Bayesian ensemble approach. Unlike standard FL methods that simply rely on averaging, FedBEns combine client models using a mixture of Laplace approximations to model multimodal local posteriors. Empirically, FEDBEns demonstrates superior performance on benchmark datasets.

**Claims And Evidence:**

The claims are generally supported by empirical evidence.

**Essential References Not Discussed:**

No critical references appear to be missing that would significantly alter the context or understanding of the proposed contributions.

**Experimental Designs Or Analyses:**

Experimental designs are sound, with clear baselines chosen from different categories of one-shot federated learning (FedFisher, RegMean, DENSE, OTFusion, FisherMerge). The analysis across varying levels of data heterogeneity and client numbers is comprehensive and methodologically robust.

**Methods And Evaluation Criteria:**

The proposed methods and evaluation criteria (datasets like FMNIST, SVHN, and CIFAR10) are appropriate and standard in federated learning literature. Experiments varying the heterogeneity parameter and client counts effectively illustrate the method’s robustness in diverse scenarios.

**Other Comments Or Suggestions:**

- Readability and clarity of the algorithm and experimental setups are very clear.

- Further exploring and clearly quantifying computational and communication trade-offs in practical deployments would strengthen the paper's applicability and appeal.

**Other Strengths And Weaknesses:**

**Strengths:**

- Clear justification for the multimodal Bayesian approach.

- Empirical results demonstrating significant accuracy improvements over baselines.

- Good evaluation across multiple datasets and varying heterogeneity levels.


**Weaknesses:**

- Computationally intensive due to multiple local models required per client, significantly impacting scalability, particularly in large-scale or resource-constrained settings.
- Quite limited novelty: The paper’s contribution is essentially an explicit application of existing Bayesian ensemble and aggregation ideas to one-shot FL without substantial conceptual innovation. If empirical results were not as strong, this lack of novelty would likely not warrant a paper.
- Lack of detailed analysis on practical constraints, such as communication overhead and latency, limiting the clear demonstration of practical applicability in real-world scenarios.

**Questions For Authors:**

- Could you further clarify the scalability and communication overhead compared to simpler baseline methods?

**Relation To Broader Scientific Literature:**

FedBEns extends the literature on Bayesian federated learning methods, particularly enhancing one-shot federated learning. It directly builds upon prior works (FedBE, FedPA, FedPop, FedHB), explicitly modeling multimodal local posteriors through Laplace mixture approximations. However, the approach is quite incremental, essentially applying known Bayesian ensemble ideas and aggregation explicitly to the one-shot federated learning setting. While simple, no other work has explicitly pursued this direction in one-shot FL, to the best of my knowledge.

**Theoretical Claims:**

This paper does not provide any new theoretical results.

---

> ### Author Rebuttal · Authors · 2025-04-01
>
> **1) Computational/Communication overhead quantification:** See answer 1) to reviewer wHXT.
>
> **2) Computational/Communication trade-offs and practical applicability:**
> The proposed method is inherently more computationally intensive, as it is based on an ensemble of M models. Its applicability depends on the specific application setting and the base model (e.g., NN architecture) used for training in the federated setting. Regarding the trade-off between computational costs and prediction performance, it is worth mentioning that our proposal substantially improves over existing methods with just a couple of mixtures (see Table 1 and Figure 2 of the manuscript).
>
> The reviewer suggested additional exploration of the computational and communication trade-offs. We provide the following insights in response.
>
> ***2.1) Comparison with FedFisher:***
> FedFisher is our direct competitor. In Appendix B.4, we extend FedFisher to a multi-round setting. By letting FedFisher train over 5 rounds, its total communication and computation costs equal those of FedBEns with 5 mixture components. Our method shows to be competitive despite relying on a single communication round. We performed an additional experiment in which we equalized the client computation with FedFisher by running FedBEns with 5 mixture components on clients’ training datasets 5 times smaller. Remarkably, FedBEns performs better than FedFisher in most scenarios (see the table below and FedFisher K-FAC results in Table 1 of the manuscript for comparison).
>
> **FedBEns, 5 mixtures Kron, REDUCED DATASETS**
> | Dataset  | Accuracy [%] (Avg ± STD, 3 seeds) |
> |-|-|
> | **Alpha = 0.05**  |  |
> | FMNIST   | 50.87 ±  4.46  |
> | SVHN   | 59,12 ± 1.42   |
> | CIFAR10 | 45.38  ± 0.62 |
> | **Alpha = 0.4**   |  |
> | FMNIST | 77.95 ± 0.31  |
> | SVHN   | 81.15  ±  0.25  |
> | CIFAR10 | 53.82  ±  1.16 |
>
> ***2.2) Other tasks:***
> To further explore the accuracy-costs tradeoff for practical applications, we conducted a more challenging experiment with a ResNet-18 on CIFAR100. We followed the experimental setting of the paper; however, we limited FedBEns to only 2 mixtures and Kronecker factorization. Local training lasted for 100 epochs. The table below shows that at the cost of doubling the computation/communications cost w.r.t. FedFisher, FedBEns achieves a higher gain in accuracy (doubled or even more).
> | Metric  | FedBEns M=2 | FedFisher | RegMean | DENSE | OTFusion | Fisher-Merge |
> |:-|-:|-:|-:|-:|-:|-:|
> | **Accuracy**, CIFAR100 (alpha=0.05) | 8.56±0.47 | 3.14±0.23 | 3.01±0.54 | 5.88±1.07 | 4.79±0.19 | 2.81±0.33 |
> | **Accuracy**, CIFAR100 (alpha=0.4)| 16.95±0.67 | 4.79±0.31 | 4.85±1.16 | 13.92±0.54 | 4.74±0.56 | 3.87±0.48 |
> | **Communication Costs (MB)** | 716 | 358 | 93 | 47 | 47 | 94 |
> | **Local runtime (min)** | 28.0 | 14.0 | 9.3 | 9.2 | 9.2 | 10.1 |
> | **Server runtime (min)** | 2.3 | 1.2 | 0.6 | 10.4 | 0.2 | 0.3 |
>
> ***2.3) System heterogeneity:***
> Another comment regarding the practical deployment of our approach is that the number of mixtures can be customized on the basis of each client's computational power. As an illustrative example, we considered CIFAR-10 with 5 clients: 3 employ a single Laplace approximation (M=1) and 2 are able to exploit three mixtures (M=3). In both cases, the final model returned to the clients is composed of a single model. The additional information coming from the clients with more computational power benefits the entire federation and leads the server to find a better global model, especially in heterogeneous settings. The table below shows the mean accuracy over five seeds for FedBEns in two settings: all clients use a single component (M=1 ALL) versus the customized setup described above.
>
> |Dataset|Accuracy [%] (Avg ± STD, 3 seeds)||
> |-|-|-|
> | | **M=1 ALL**| **M CUSTOMIZED**|
> |**Alpha = 0.05**| | |
> |FMNIST| 57.02 ± 4.73| 59.97 ± 1.44|
> |SVHN| 66.11 ± 0.77| 68.39 ± 1.57|
> |CIFAR10| 39.89 ± 1.71| 42.32 ± 2.25|
> |**Alpha = 0.4**| | |
> |FMNIST| 77.07 ± 0.29|78.31 ± 1.76|
> |SVHN| 82.86 ± 0.21|82.75 ± 1.41|
> |CIFAR10| 53.97 ± 0.86|54.68 ± 1.88|
>
>
>  In the case of acceptance, we will integrate these additional results into the paper.
>
> **3) Quite limited novelty:**
> We agree with the reviewer that our proposal relies on concepts commonly used in the Bayesian learning landscape. However, we would like to emphasize that our key contribution lies in recognizing how these ideas can significantly improve FL, specifically in a One-Shot setting. In particular, we highlight the importance of explicitly capturing different modes of the posterior rather than framing the FL task purely as an optimization problem. Additionally, we believe that, by building upon a well-studied theoretical framework, our method, also thanks to its inherent simplicity, is both principled and robust. Moreover it brings, as pointed out by the reviewer, very strong empirical results.

---

> > ### Comment · Reviewer_dxFK · 2025-04-06
> >
> > Thanks for your rebuttal.
> >
> > Although undeniable that your method gets good results against one-shot methods, the fact that its communication and computation costs are much more similar to "M-shot methods", means that I am really struggling to view the respective benchmarks given in the main body of the paper as fair comparisons. I am less impressed with the numerical results than I was after the initial review.
> >
> > Regarding your response to **Computational/Communication trade-offs and practical applicability**, I can understand this, but this is like reasoning that M-shot methods are worth it over one-shot methods; except in this instance, the method is being *proposed* as a one-shot method.
> >
> > I will keep my score as is, but am hesitant to recommend acceptance of the paper given my current understanding of the algorithm complexity and its comparison to existing literature.

---

> > > ### Author Response · Authors · 2025-04-09
> > >
> > > We thank the reviewer for engaging in a discussion and providing feedback.
> > >
> > > In the following, we provide some comments hoping to clarify the concerns regarding: 1) Fairness of the comparison with SOTA One-Shot methods; 2) Why a One-Shot approach is worthwhile, even with comparable client costs to Few-Shots methods.
> > >
> > > **1) Comparison Fairness**
> > >
> > > We emphasize that also the evaluated baselines differ in communication and computation costs, with more resource-intensive methods generally performing better (see first table to reviewer wHXT’s rebuttal and Tab.1 in the paper). For example, FedFisher transmits roughly 5x more data than Dense and we have checked that a 5-shot Dense outperforms FedFisher in terms of accuracy (up to 11% on CIFAR10, 5 clients, $\alpha=0.1$). We do not think this would be a valid reason to blame FedFisher.
> > >
> > > In addition, the fairness of a comparison between One-Shot and Few-Shot approaches can be debatable, as they rely on fundamentally different assumptions: One-Shot methods operate without server’s feedback, while multi-round approaches exploit it to refine the global model. Thus, a direct comparison based solely on resource consumption overlooks this distinction.
> > >
> > > Our comparison with multi-round FedFisher was mainly performed to check whether, even in a setting that inherently advantages multi-round approaches, our method gives competitive results, *despite* being a One-Shot approach. The positive outcome even surprised us, as we expected FedFisher to perform significantly better in this scenario.
> > >
> > > We acknowledge that FedBEns can be costly, particularly with many mixtures. However, we regard the number of mixtures as a tunable parameter to balance the performance-cost trade-off in specific application scenarios. The accuracy of FedBEns with a single mixture (M=1) is comparable to, or slightly better than, the baselines across all datasets and heterogeneity levels (compare Fig. 2 and Tab. 1 in the paper), and FedBEns significantly outperforms the baselines already for M=2. In terms of computational/communication costs (see the first table in the rebuttal to reviewer wHXT), FedBEns (M=1) is comparable to FedFisher and only marginally more demanding than RegMean, the best performance methods aside from FedBEns.
> > >
> > > **2) Advantages of a One-Shot approach vs Few-Shots even when client costs are comparable**
> > >
> > > While FedBEns with M mixtures and FedFisher with M rounds entail similar per-client communication/computation costs, the overall time a client needs to participate to the training may differ significantly due to 1) server-to-client transmissions, 2) the straggler effect. After discussing these issues, we comment on the importance of a shorter client participation time.
> > >
> > > *Server-to-client transmissions.* Assuming equal server/clients uplink and downlink
> > > capacities, the communication time for M-round FedFisher scales as 2M, since the model must be sent from client to server and back in each round. In contrast, FedBEns communication time is halved since mixtures are sent once and then aggregated. If communication is the bottleneck (as is often the case in FL systems), training M-round FedFisher would require each client to remain available for roughly twice as long as FedBEns with M mixtures. (Note that FedBEns final model does not change if a client leaves after having sent its mixtures).
> > >
> > > *Straggler effect.* The reasoning above assumes homogeneous clients and synchronization. However, real-world systems are affected by stragglers, clients whose updates arrive significantly later than those of others [1]. Stragglers slow down the entire system, as the server must wait for their inputs before generating a new model to distribute. This, in turn, forces other clients to wait for the updated model, further increasing their participation time. In the case of one-shot algorithms, clients can leave as soon as they provide their contributions.
> > >
> > > *Importance of a shorter participation time.* In a cross-device setting, clients often exhibit volatile participation patterns. Therefore, minimizing the required client's participation time is critical: it may determine whether a client contributes to the training or drops out before completion, thereby preventing the federation from leveraging its dataset and degrading the final model’s quality [1]. Finally, we observe that in FedBEns, each client can transmit each mixture as soon as it is computed. As a result, even if a client disconnects before completing all M mixtures, the system benefits from its partial contribution (on an additional experiment in line with those of the first rebuttal, ‘System heterogeneity’ section, 2 clients with 3 mixtures each lose up to 4% accuracy vs 5 clients with 1 mixture, and up to 7% vs the customized scenario).
> > >
> > > We thank again the reviewer, we will expand this discussion in the paper if accepted
> > >
> > > [1] Kairouz, et al. "Advances and open problems in federated learning." Foundations and trends in machine learning, ‘21

---

### Official Review · Reviewer_b5tc · 2025-03-10

**Overall Recommendation:** 3

**Summary:**

This paper focuses on one-shot Federated Learning, where the model is aggregated in a single communication round. The authors provide an analysis through the lens of Bayesian inference and then propose a method to leverage the inherent multimodality of local loss functions to find better global models.

**Claims And Evidence:**

na

**Essential References Not Discussed:**

na

**Experimental Designs Or Analyses:**

na

**Methods And Evaluation Criteria:**

na

**Other Comments Or Suggestions:**

na

**Other Strengths And Weaknesses:**

Overall, the results look promising. However, it would be beneficial to include discussions on the following aspects:

- computation: The algorithm involves additional server-side training. It would be fair to compare its computational cost with other methods, as some model fusion methods don't require additional training.

-  Figure 1 is not very clear or easy to understand. For example, where is the global loss in the left figure? Are both figures plotted in the same way? Clarifying these points would improve readability.

-  It would be interesting to see whether this method converges faster when trained on the server.

**Questions For Authors:**

na

**Relation To Broader Scientific Literature:**

na

**Theoretical Claims:**

na

---

> ### Author Rebuttal · Authors · 2025-04-01
>
> **1) Additional server-side computation:**
> We appreciate the reviewer’s observation—this is indeed an important point. Our method does entail increased server-side computation as the number of mixture components grows. However, we would also like to clarify that some of the baselines considered can be similarly demanding in terms of server-side resources. For instance, DENSE involves both synthetic data generation and knowledge distillation at the server, which results in a training time comparable to that of FedBEns with 5 mixture components.
> Furthermore, we would like to highlight that FedBEns outperforms DENSE even when using fewer mixture components (compare the results for FedBEns in Figure 2 with those in Table 1 in the manuscript) while requiring less server-side training time. To illustrate this more concretely, we provide below a table—also to be included in the paper—that reports the server training time for the CIFAR-10 experiment with 5 clients. The number of mixture components used in FedBEns is denoted as M.
>
> |Method|Server average execution time (seconds)|
> |-|-|
> |FedBEns (Kron)|M=5 173.5, M=4: 113.5, M=3: 65.9, M=2: 28.5, M=1: 7.1|
> |DENSE|170.2|
> |FedFisher|13.4|
> |Regmean|5.1|
> |OTFusion|1.1|
> |FisherMerge|0.9|
>
> Our approach increases server computation but improves predictive performance. Whether this additional computational cost is worth it depends on the practical context, and it can be justified if the server has enough computational power.
>
> For an additional discussion on the computational/communication cost, please refer to answer 1) to reviewer wHXT.
>
> **2) Figure 1, clarification:**
> The two subfigures in Figure 1 illustrate different functions, as also indicated by the respective titles at the top of each subfigure. The left plot shows the local loss functions for client 1 (orange curves) and client 2 (blue curves), each with its own optimum (represented by orange and blue points, respectively). The right plot depicts the global loss function (green curve), obtained using Eq. (2) by combining the losses of the two clients from the left plot. The key takeaway is that, when estimating the global loss, capturing the secondary optimum is more crucial than the primary one, emphasizing the need for a multi-modal approach. We will modify the figure caption to make it clearer.
>
> **3) Faster convergence when trained on the server?:**
> If we have correctly understood the reviewer’s point, our method can indeed also be used in a centralized setting—that is, where all training data is available on the server. However, in this case, the server’s role in merging different posteriors becomes redundant, as there is no need to combine client-specific posteriors. Naturally, centralized training also eliminates communication overhead, providing an additional efficiency advantage. On the other hand, the federated approach could benefit from the inherent parallelism of distributed data processing. Overall, the relative training speed between centralized and federated setups depends on the interplay of these three factors, as well as potential differences in the hardware available in each case.
> As an illustrative example, in a centralized setting and using the same hardware described in Sec. 5.3, on CIFAR10, computing one mixture component over the entire dataset would require 129.5 seconds for training the Neural Network, plus 25 seconds for computing the Hessian. In a federated setting, the time to compute a client’s mixture component would decrease proportionally with the number of clients, due to parallelism. However, aggregating the results incurs a cost that scales linearly with the number of clients, as computing the global log-posterior in Eq. (4) requires summing over the clients' posteriors. Assuming identical hardware at the server and at the clients and ignoring communication delays, the federated approach on CIFAR10 and 5 mixtures is more efficient up to $\sim 20$ clients. This threshold decreases when communication overhead is included or when the centralized server has superior computational capabilities.
>
> Last, also in a centralized setting, accuracy increases with mixtures. As an example we report the average accuracy (3 seeds) for CIFAR10 as a function of the number of mixtures M, with the same training details reported in the paper: M=1: 79.48%, M=2: 82.11%, M=3: 83.28%, M=4: 84.99%, M=5: 85.46%.

---

### Official Review · Reviewer_wHXT · 2025-03-16

**Overall Recommendation:** 4

**Summary:**

The paper introduces FedBEns, a one-shot federated learning (FL) algorithm using Bayesian inference to address multimodal local loss functions. It approximates local posteriors with a mixture of Laplace approximations (GMM) and aggregates them to estimate the global posterior. The server identifies global modes via SGD and performs ensemble predictions. Experiments on FashionMNIST, SVHN, and CIFAR10 show FedBEns outperforms baselines (e.g., FedFisher, RegMean) by up to 10% in accuracy, especially under high data heterogeneity (α=0.05).

**Claims And Evidence:**

Employing GMMs to model multi-modality of local loss for each client is straightforward and the proposed method is solid. Extensive numerical experiments support its validity.

**Essential References Not Discussed:**

Please refer to "Relation to Broader Scientific Literature" Section

**Experimental Designs Or Analyses:**

Experiments (Tables 1-4) vary heterogeneity and client numbers, with 5 seeds for robustness. The ablation studies on mixture size, Hessian approximations, temperature, and prior variance (Figures 2-5) are sound. especially, an ablative study for the number of mixture components validates the superiority of employing multimodality instead of unimodal gaussian. However, the lack of statistical significance tests (e.g., t-tests) for accuracy comparisons may limit the reliability of superiority claims. In addition, the assumption of having validation data at the server is unrealistic in federated learning (FL), as FL typically assumes no raw training data exists at the server. If such data is available, it should be used for training rather than just validation.

**Methods And Evaluation Criteria:**

The method uses a mixture of Laplace approximations, which aligns with the goal of capturing multimodal posteriors in one-shot FL. Evaluation on standard datasets (FashionMNIST, SVHN, CIFAR10) with Dirichlet sampling for heterogeneity is appropriate.

**Other Comments Or Suggestions:**

- Validation Set Usage: Why use a server-side validation set for hyperparameter tuning, given FL’s privacy constraints? If removed, how would performance change?

- Computational Cost: Can you quantify the runtime and communication costs of FedBEns compared to baselines?

- Does the proposed method work under feature hetergeneity such as domain shift?

**Other Strengths And Weaknesses:**

The claim of computational efficiency is weakly supported—while a trade-off with mixture size is mentioned, no runtime or communication cost analysis is provided.

**Questions For Authors:**

- Learnable Mixture Weights: Is it possible to make each mixture component weight learnable?

**Relation To Broader Scientific Literature:**

FedBEns builds on Bayesian FL by focusing on one-shot learning and multimodal posteriors, unlike FedPA’s multi-round MCMC approach. It extends Laplace approximations (MacKay, 1992) to mixtures, similar to the work in Eschenhagen et al. (2021). It contrasts with unimodal loss-based methods like FedFisher (Jhunjhunwala et al., 2024), emphasizing multimodal loss landscapes.

**Theoretical Claims:**

Proposition 3.1 (Equation 2) claims the global posterior can be derived by combining local posteriors under conditional independence and same-prior assumptions. The proof (Appendix A) is correct, using Bayes’ theorem and factorization of likelihoods. Extension to unimodal to multimodal posteriors is straght-forward.

---

> ### Author Rebuttal · Authors · 2025-04-01
>
> We are happy the reviewer appreciated the paper.
>
> **1) Runtime/communication costs quantification:**
> We observe that the FedBEns per-client computational and communication costs are both linear in the number of mixtures M. On the contrary, in our implementation, the computation time at the server scales as M$^2$ (we have to find M modes of the global log-posterior through M SGD runs, and the cost of each evaluation of the global log-posterior scales linearly with M, see Eq. (4) ).
> We provide experimental results for the CIFAR10 dataset with 5 clients on the hardware specified in Sec. 5.3. Times are averaged over 5 runs.
> - Local training time: for each mixture component, 25.9s for training the NN and 5.0s for computing the Hessian with Kronecker factorization.
> - Communication cost: for each mixture component, 17.8 MB (3.6 MB for the NN and 14.2 for the Hessian with Kronecker factorization).
> - Server aggregation time: ($O(M^2)$). We report the execution time for different values of M: M=1: 7.1s, M=2: 28.5s, M=3: 65.9s, M=4: 113.5s, M=5: 173.5s.
>
> We provide the communication and computation costs with the other baselines in the following table. For additional comments on this aspect and results comparing FedBEns and FedFisher under similar computational cost, see answer 2) to reviewer dxFK:
>
> |Method|Client runtime (s)|Server runtime (s)|Communication cost (MB)|
> |-|-|-|-|
> |FedBEns(kron)|30.9$\cdot$M|7.1$\cdot$M$^2$|17.8$\cdot$M|
> |FedFisher-KFAC|30.9|13.4|17.8|
> |RegMean|26.0|5.1|7.3|
> |FisherMerge|26.2|0.9|7.31|
> |DENSE|25.9|170.2|3.65|
> |OTFusion|25.9|1.1|3.65|
>
> **2) Validation set usage:**
> We thank the reviewer for the interesting question. First, we note that FedFisher, RegMean, DENSE, OTFusion rely on a validation set at the server, both in their original papers and in our experiments. Our method uses a validation set during the server's gradient descent runs, to find the parameter configuration that achieves the best validation performance (Sec. 5.3). We have performed an additional experiment where no validation dataset is used and the number of server epochs was fixed to 100. Remarkably, FedBEns still outperforms the considered baselines, which continue to take advantage of the validation dataset, with only a minor drop in accuracy (up to a few p.p.; see Table 1 in the paper and the supplementary table below). We will highlight this important point more clearly in the revised version.
>
> **FedBEns, 5 mixtures Kron, NO VALIDATION**
> |Dataset| Accuracy [%] (Average, 3 seeds)|
> |-|-|
> |**Alpha = 0.05**||
> |FMNIST|59.15|
> |SVHN| 70.26|
> |CIFAR10|48.81|
> |**Alpha = 0.4**||
> |FMNIST|82.16|
> |SVHN|84.69|
> |CIFAR10|60.00|
>
> **3) Feature heterogeneity:**
> We are not sure how to interpret the expression “feature heterogeneity.” If it refers to structurally different data across clients—e.g., clients having different numbers or types of features—this would require a different model architecture per client. Since our training method assumes a shared model architecture across all clients, it is not directly applicable to such a setting. In contrast, our method explicitly assumes *statistical heterogeneity* across client datasets—that is, data drawn from different underlying distributions. We do not impose strong assumptions on these distributions beyond conditional independence. If the reviewer refers to *domain shifts* after global model training, our method remains applicable in certain scenarios, such as covariate shift. In fact, FedBEns can be particularly beneficial in this context: when new clients join the federation with different data distributions, the approximate posterior produced by FedBEns can serve as a prior. This acts as a regularizer, helping to prevent the model from drifting too far when fine-tuned on data that differs significantly from the original training distribution. We believe that exploring this aspect more thoroughly would be an interesting direction for future work.
>
> **4) Learnable Weights:**
> A principled way to assign the weights would be to set them equal to the model evidence of each Gaussian component. We experimented with this approach and found that performance remained essentially unchanged, as the model’s evidences were comparable to each other, in line with what has been found in [1]. Another possibility would be to let the server weigh each distribution, or all distributions belonging to a given client, based on the predictive performance of the model trained by that client. However, this approach may require the server to have explicit access to a dataset while, in our approach, it is not strictly needed (as discussed also in answer 2).
>
> [1] Immer et al. “Scalable marginal likelihood estimation for model selection in deep learning”, ICML ‘21
>
> **5) Statistical Significance:** all results (Table 1 and 2) are statistically significant, p-value<0.05, computed with a paired t-test between our models and the best-performing competitor.

---

### Decision · Program_Chairs · 2025-05-01

**Decision:**

Accept (poster)

**Comment:**

This paper uses a mixture of (Laplace) Gaussians in one-shot federated learning. Reviewers agree that the multimodal nature of local posteriors is well-motivated, that the method itself is straightforward, and that the experiments are performed well (good benchmarks and ablations). Concerns such as computation / communication cost, and some additional experiments, were addressed by the authors during rebuttal. The authors have committed to adding these results and discussions into the paper. Overall, this is a solid contribution to ICML that takes well-known methods and then applies them to a new setting, with good motivation and experiments.

Reviewer dxFK asked about the benefit of one-shot federated learning (but M times the cost) vs M shots. I agree with the authors that this is still an interesting and important setting.